Journal of Machine Learning Research 23 (2025) 1-13          Submitted 17/06/25; Published

# Uncertainty-Aware Ensemble Segmentation of Breast Cancer Tissue Microarrays

**Lucía Schmidt-Santiago**[*]        LSCHMIDT@PA.UC3M.ES

**Roman Kinakh**[*†]        RKINAKH@ING.UC3M.ES

**Sergio Carreras-Salinas**        SCARRERA@PA.UC3M.ES

**Sara Guerrero-Aspizua**        SGUERRER@ING.UC3M.ES

**Gonzalo R. Ríos-Muñoz**        GRIOS@ING.UC3M.ES

**Arrate Muñoz-Barrutia**        MAMUNOZB@ING.UC3M.ES

*Bioengineering Department*

*Universidad Carlos III de Madrid, Spain*

*Legánes, Madrid, 28911, Spain*

**Editor:** -

## Abstract

Breast cancer Tissue Microarrays (TMAs) offer a high-throughput platform for studying tumor morphology and biomarker expression. We present an automated deep learning pipeline for semantic segmentation of Hematoxylin and Eosin (H&E)-stained breast cancer TMAs, integrating ensemble U-Net architectures with ResNet encoders and Monte Carlo Dropout (MCDO) for uncertainty estimation. A robust pre-processing workflow addresses illumination artifacts, staining variability, and tissue detection.

Multiple U-Net models were trained using distinct loss functions to address class imbalance and feature diversity. Predictions were combined via soft voting, emulating consensus among pathologists. Uncertainty was quantified using MCDO across ensemble outputs, enhancing reliability and interpretability.

Our pipeline outperforms similar methods such as WeGleNet (mIoU = 0.4368) and HistoSegNet (mIoU = 0.5505), achieving a mean IoU of $0.58 \pm 0.11$ and Dice Score of $0.66 \pm 0.10$. Calibration analysis shows superior alignment of standard deviation–based uncertainty estimates with actual prediction errors (UCE = $0.085 \pm 0.033$). This pipeline effectively segments complex histopathological structures and flags ambiguous regions for review, supporting downstream biomarker discovery and clinical interpretation.

**Keywords:** tissue microarrays, deep learning, segmentation, ensemble, uncertainty, histopathology.

## 1 Introduction

Breast cancer accounts for 12.5% of all cancer diagnoses worldwide, with over 2.3 million new cases and 685,000 deaths annually (Ferlay et al. (2021)). Its clinical complexity arises from pronounced biological heterogeneity and the dynamic tumor microenvironment (TME), which influences disease progression and treatment response.

---

*. These authors contributed equally to this work.

†. Corresponding author: `rkinakh@ing.uc3m.es`

Tissue Microarrays (TMAs) have become essential tools in cancer research, enabling high-throughput analysis of tumor morphology and biomarker expression from multiple samples (Nocito et al. (2001)). However, their densely packed layout, staining variability, and tissue heterogeneity complicate automated analysis. Reliable segmentation of histopathological regions—such as tumor epithelium (TUM), tumor-associated stroma (STR), necrosis (NEC), and inflammation (INF)—is vital for downstream clinical and translational applications, especially in HER2-positive (HER2+) and Tripple Negative Breast Cancer (TNBC) (Salgado et al. (2015); Dieci et al. (2015)).

Deep learning has significantly advanced digital pathology. The U-Net architecture (Ronneberger et al. (2015)) is widely adopted for medical image segmentation due to its strong localization and contextual learning abilities. However, challenges remain, including class imbalance, staining inconsistencies, and ambiguous tissue boundaries (Van Eycke et al. (2017)). These issues especially affect performance on underrepresented tissue types.

To improve robustness, ensemble learning aggregates outputs from models trained with diverse loss functions, reducing individual biases and improving generalization (Kamnitsas et al. (2017)). Additionally, uncertainty estimation techniques such as Monte Carlo Dropout (MCDO) (Gal and Ghahramani (2016b)) have proven effective in highlighting ambiguous predictions, supporting model interpretability and trustworthiness in clinical workflows (Kendall et al. (2017)).

In this study, we propose a pipeline for breast cancer TMA segmentation that combines pre-processing, an ensemble of U-Net models with ResNet encoders, and MCDO-based uncertainty estimation. This approach addresses staining variability, class imbalance, and model interpretability. Our goal is to deliver accurate, interpretable segmentations to support biomarker discovery and deepen understanding of tumor biology.

## 2 Material and Methods

### 2.1 Dataset

We used a publicly available dataset from the British Columbia Cancer Agency (BCCA), comprising Hematoxylin and Eosin (H&E)-stained TMA cores from 4,944 breast cancer patients (Genetic Pathology Evaluation Centre). Each patient contributed three digitized 0.6 mm cores at a resolution of 2256×1440 pixels.

For this study, we focused on 123 manually annotated TMA cores with pixel-wise labels, obtained through a three-step annotation process: an expert pathologist identified regions of interest, and a trained assistant delineated TUM, NEC, and INF using QuPath (Bankhead et al. (2017)), which was then evaluated by the former. These three classes were prioritized due to their prognostic significance in breast cancer and their relevance in downstream biomarker and immune infiltration analyses. 110 images were used for training and validation while the remaining 13 images were reserved for testing. The random selection was performed based on patient name identifiers such that no images from the same patient appeared in both the test and non-test sets.

The miscellaneous (MIS) class, consisting primarily of background, non-cellular and disrupted tissue regions, was automatically segmented in the first step using simple thresholding and morphological operations (Figure. 1), as these areas are typically easy to isolate due to their predominantly white appearance. Following this, the STR class was derived by

subtracting the manually annotated TUM, NEC, and inflammation/tumor-infiltrating lymphocytes (INF/TILs) regions from the remaining tissue, excluding areas already assigned to the MIS class.

The final dataset contains 123 annotated cores: TUM (n=123), STR (n=123), NEC (n=47), INF (n=52), and MIS (n=123). All images were square-cropped to the edges of the core and resized to 1024×1024 pixels resulting in a uniform pixel size of $\approx 0.6\mu m$.

## 2.2 Pre-processing and Data Augmentation

Comprehensive pre-processing was implemented to enhance the model's training capacity by addressing variability in tissue sections and staining. As shown in Fig. 1, the pipeline included illumination correction, tissue segmentation in the LAB color space, morphological post-processing, and contour filtering to exclude noise and small objects, and finally square cropping centered on the primary TMA object. To address staining inconsistencies, Reinhard et al.'s Stain Color Normalization (SN) method (Reinhard et al. (2001)) aligned color distributions, enhancing dataset uniformity while preserving morphological details.

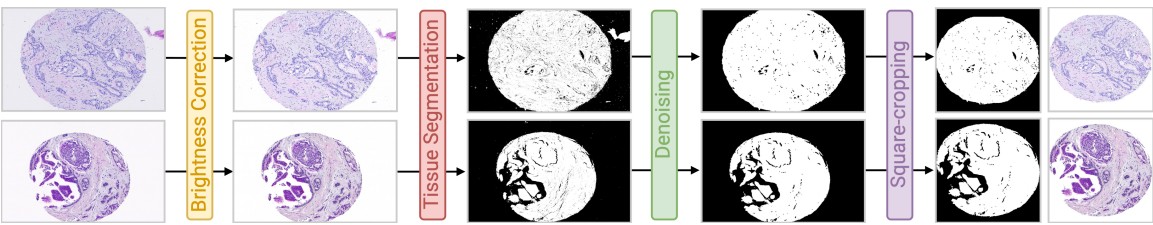

Figure 1: **Pre-processing pipeline.** From left to right, the process begins with brightness correction to improve TMA core image contrast, followed by segmentation to distinguish tissue from the background. Post-processing steps include Salt & Pepper noise reduction and morphological operations to refine tissue boundaries. Finally, square cropping is applied around the main tissue object.

Due to the sparsity of manually annotated data, we employed a data augmentation strategy that introduces realistic variations into the training dataset (Tellez et al. (2019)) to enhance the generalization ability of our models and improve their robustness. Our augmentation pipeline includes random geometrical transformations such as rotations, vertical and horizontal flips, along with color adjustments, including changes in brightness and contrast. These modifications were applied with a probability of $p = 0.2$.

## 2.3 Model Architecture

Our segmentation model combines a U-Net architecture (Ronneberger et al. (2015)) with a ResNet50 encoder (He et al. (2016)), leveraging both spatial precision and deep feature extraction. As shown in Fig. 2, the encoder, pre-trained on ImageNet (Deng et al. (2009)), extracts hierarchical features from input images $\mathbf{X} \in \mathbb{R}^{C \times H \times W}$, while skip connections preserve spatial details.

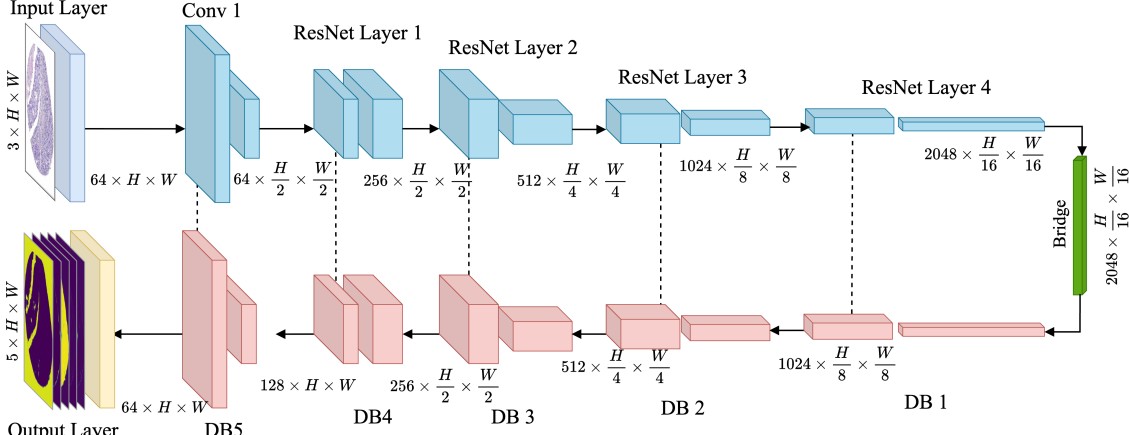

Figure 2: **Architecture of the U-Net with ResNet50 encoder.** The encoder (blue) consists of a Conv1 layer and ResNet50 bottleneck blocks for efficient feature extraction and dimensionality reduction, with skip connections (dashed lines) to the decoder. The bridge (green) links the encoder and decoder, while the Decoder Blocks (DB, red) upsample, integrate skip connections, and refine feature maps. The final layer applies a $1 \times 1$ convolution and $softmax$ to generate class probability maps. $H$ and $W$ represent the height and width of the input image.

A bridge module refines deep features before decoding. The decoder upsamples and fuses features with encoder outputs to reconstruct the spatial layout. The final layer applies a $1 \times 1$ convolution followed by $softmax$ to produce class probability maps. The predicted class for each pixel $i$ is given by: $\hat{y}_i = \arg\max_c \left( \frac{\exp(z_{i,c})}{\sum_{k=1}^{n} \exp(z_{i,k})} \right)$, where $z_{i,c}$ is the logit for class $c$ and $n$ is the total number of classes.

## 2.4 Monte Carlo Dropout

To quantify model confidence, we employ MCDO, which approximates epistemic uncertainty by performing multiple stochastic forward passes with dropout at inference time (Gal and Ghahramani (2016a)). This produces a predictive distribution for each pixel from which uncertainty metrics are derived.

We compute two measures: standard deviation (STD), which captures variation across predictions (see Appendix B), and Entropy (H), which reflects the unpredictability of the mean prediction for each pixel. For pixel $i$, entropy is defined as:

$$H_i = -\sum_{c=1}^{C} \left( \bar{p}_{i,c} \log_2 \left( \bar{\sigma}_{i,c} + \epsilon \right) + (1 - \bar{p}_{i,c}) \log_2 \left( 1 - \bar{p}_{i,c} + \epsilon \right) \right),$$

where $C$ is the number of classes, $\bar{p}_{i,c}$ and $\bar{\sigma}_{i,c}$ are the mean and standard deviation of predicted probabilities for class $c$ at pixel $i$, and $\epsilon$ ensures numerical stability.

Both uncertainty maps are normalized to [0, 1] for visualization and calibration.

## 2.5 Training and Loss Functions

Multiple U-Net models were trained on an Nvidia RTX 3090 GPU (24 GB VRAM) using 5-fold cross-validation. Each fold used a batch size of 3, a learning rate of $10^{-4}$, and 50 epochs. The Adam optimizer was used, and the best model per fold was selected based on validation IoU. Final evaluation was performed on 13 manually annotated TMA cores. For MCDO, a dropout rate of 0.3 was applied during inference.

To address class imbalance and segmentation ambiguity, we trained models with five loss functions: Cross-Entropy, Dice, Focal, Focal Dice, and Tversky (Appendix A). Each emphasizes different properties of the prediction error, from probabilistic confidence to region overlap.

## 2.6 Ensemble Model

Given the diverse and complex nature of the dataset, a soft voting mechanism is employed to ensemble outputs from models trained with different loss functions. This approach aggregates the predicted class probabilities for each pixel across all models and selects the class with the highest cumulative probability as the final prediction: $C^* = \arg\max_c \left( \sum_{m=1}^{M} P_m(c \mid p) \right)$, where $C^*$ is the final predicted class for pixel $p$, $c$ represents a class label, $M$ is the total number of models, and $P_m(c \mid p)$ is the predicted probability for class $c$ at pixel $p$ from model $m$. This approach not only ensures a more stable and accurate segmentation outcome but also mimics the collaborative decision-making process often employed by pathologists in clinical settings.

## 2.7 Evaluation

We evaluated segmentation performance using standard pixel- and region-level metrics, including Intersection over Union (IoU), Dice Score, Accuracy, Precision, and Recall. Definitions of these are provided in the Appendix C.

Two additional metrics are introduced to account for clinical relevance:

**Frequency-Weighted IoU (fw-IoU)** weighs IoU values by class frequency: fw-IoU = $\sum_{c=1}^{N} w_c \cdot \text{IoU}_c$, where $w_c = |y_c| / \sum_{i=1}^{N} |y_i|$.

**Frequency–relevance–weighted IoU.** We report frw-IoU defined inline as frw-IoU = $\sum_c p_c \, w_c \, \text{IoU}_c$, extending fw-IoU with clinical relevance weights $p_c$ ($\sum_c p_c = 1$). We set $p_c = \{\text{MIS} : 0.05, \text{STR} : 0.15, \text{NEC} : 0.15, \text{INF} : 0.15, \text{TUM} : 0.50\}$ to reflect asymmetric utility: TUM is clinically paramount and anchors downstream readouts; NEC is an adverse prognostic feature; stromal proportion (STR; tumor–stroma ratio) is prognostic; and tumor-infiltrating lymphocytes (INF/TILs) are prognostic/predictive, especially in TNBC/HER2+; MIS is minimally informative (Chen et al., 2023; Yan et al., 2022; Salgado et al., 2015; Dieci et al., 2015). This task-aligned weighting is consistent with recommendations for clinically meaningful validation (Maier-Hein et al., 2024).

To evaluate uncertainty quality, we computed the **Uncertainty Calibration Error (UCE)**, which measures alignment between uncertainty estimates and actual errors: UCE = $\sum_{b=1}^{B} w_b \cdot |\text{ErrRate}(b) - \hat{u}(b)|$, where $w_b$ is the proportion of samples in bin $b$, and $\hat{u}(b)$ is the mean uncertainty in that bin. We report UCE for both entropy- and STD-based uncertainty maps. Reliability diagrams (Fig. 4) illustrate calibration quality.

An ablation study was conducted to evaluate the effects of SN and MCDO. Four configurations (with/without SN and MCDO) were tested, and the mean IoU was further computed per class.

## 3 Results

### 3.1 Ablation

We conducted an ablation study to assess the effects of stain SN and MCDO on segmentation performance (Table 1). SN preserved structural integrity, yielding an average Structural Similarity (SSIM) of 0.95 (Wang et al. (2004)). The ensemble without SN or MCDO achieved the highest performance (mIoU = $0.58 \pm 0.11$, Dice = $0.66 \pm 0.10$, fw-IoU = $0.80 \pm 0.07$), and was selected for visualization and benchmark comparison.

Incorporating MCDO substantially reduced segmentation metrics but improved interpretability due to uncertainty estimation. In this configuration, interestingly, SN slightly improves the performance. The configuration with both MCDO and SN (mIoU = $0.52 \pm 0.07$) offered a good balance and was chosen for uncertainty estimation (see Section 3.3).

We also compared loss functions individually and as part of an ensemble. The ensemble did not outperform standalone models across multiple metrics. However, it consistently achieved lower Standard Deviation ($\sigma$) with Means ($\mu$) almost as high as in the best performing standalone models, indicating a more consistent performance across different images (Table 2) and delivered stable results across all tissue classes (Table 3).

Table 1: Impact of MCDO and SN on segmentation metrics ($\mu \pm \sigma$). Bold indicates the best results.

| MCDO | SN | IoU | Dice | Accuracy | Precision | Recall | frw-IoU | fw-IoU |
|---|---|---|---|---|---|---|---|---|
| – | – | **0.58 ± 0.11** | **0.66 ± 0.10** | **0.87 ± 0.05** | **0.69 ± 0.09** | **0.68 ± 0.10** | **0.73 ± 0.10** | **0.80 ± 0.07** |
| – | ✓ | 0.50 ± 0.10 | 0.57 ± 0.11 | 0.81 ± 0.15 | 0.62 ± 0.09 | 0.58 ± 0.10 | 0.63 ± 0.23 | 0.74 ± 0.14 |
| ✓ | – | 0.50 ± 0.08 | 0.57 ± 0.08 | 0.87 ± 0.07 | 0.59 ± 0.08 | 0.59 ± 0.07 | 0.73 ± 0.11 | 0.79 ± 0.09 |
| ✓ | ✓ | 0.52 ± 0.07 | 0.59 ± 0.07 | **0.87 ± 0.05** | 0.61 ± 0.08 | 0.61 ± 0.07 | 0.75 ± 0.09 | **0.80 ± 0.07** |

Table 2: Segmentation metrics ($\mu \pm \sigma$) across loss functions. Bold indicates the best results.

| Loss | IoU | Dice | Accuracy | Precision | Recall | frw-IoU | fw-IoU |
|---|---|---|---|---|---|---|---|
| Cross-Entropy | **0.59 ± 0.14** | **0.67 ± 0.14** | 0.86 ± 0.05 | 0.70 ± 0.12 | **0.70 ± 0.14** | 0.71 ± 0.09 | 0.78 ± 0.07 |
| Dice | 0.55 ± 0.07 | 0.62 ± 0.07 | 0.87 ± 0.04 | 0.65 ± 0.08 | 0.64 ± 0.07 | **0.73 ± 0.10** | **0.80 ± 0.06** |
| Focal | 0.59 ± 0.12 | 0.66 ± 0.12 | 0.85 ± 0.10 | **0.72 ± 0.12** | 0.68 ± 0.11 | 0.69 ± 0.16 | 0.77 ± 0.10 |
| Focal-Dice | 0.57 ± 0.13 | 0.64 ± 0.13 | 0.86 ± 0.07 | 0.67 ± 0.11 | 0.66 ± 0.12 | 0.70 ± 0.11 | 0.78 ± 0.08 |
| Tversky | 0.53 ± 0.08 | 0.60 ± 0.08 | **0.87 ± 0.05** | 0.65 ± 0.08 | 0.62 ± 0.07 | 0.72 ± 0.09 | 0.79 ± 0.06 |
| **Ensemble** | 0.58 ± 0.11 | 0.66 ± 0.10 | **0.87 ± 0.05** | 0.69 ± 0.09 | 0.68 ± 0.10 | **0.73 ± 0.10** | 0.80 ± 0.07 |

Table 3: Per-class IoU ($\mu \pm \sigma$) across loss functions. Bold indicates the best results.

| Loss | MIS | STR | TUM | NEC | INF |
|---|---|---|---|---|---|
| Cross-Entropy | 0.93 ± 0.03 | 0.64 ± 0.20 | 0.64 ± 0.17 | 0.14 ± 0.31 | 0.14 ± 0.21 |
| Dice | 0.93 ± 0.03 | 0.65 ± 0.18 | **0.68 ± 0.18** | **0.24 ± 0.34** | 0.11 ± 0.20 |
| Focal | 0.94 ± 0.03 | **0.66 ± 0.19** | 0.62 ± 0.22 | 0.14 ± 0.30 | **0.15 ± 0.23** |
| Focal-Dice | 0.94 ± 0.03 | 0.65 ± 0.20 | 0.64 ± 0.19 | 0.18 ± 0.30 | 0.14 ± 0.22 |
| Tversky | 0.94 ± 0.03 | 0.65 ± 0.20 | 0.66 ± 0.19 | 0.17 ± 0.30 | 0.09 ± 0.19 |
| **Ensemble** | **0.94 ± 0.03** | **0.66 ± 0.19** | 0.67 ± 0.18 | 0.18 ± 0.31 | 0.13 ± 0.21 |

### 3.2 Final Ensemble

Table 2 reports segmentation performance across loss functions and the ensemble approach, while Table 3 details per-class IoU values. Although the ensemble does not lead in all standard metrics (mIoU $= 0.58 \pm 0.11$, Dice $= 0.66 \pm 0.10$), it achieves the highest scores on weighted metrics (frw-IoU $= 0.73 \pm 0.10$, fw-IoU $= 0.80 \pm 0.07$), indicating better overall balance and generalization. Its strong performance on the TUM class (mean IoU $= 0.67$) emphasizes clinical relevance.

While Dice and Focal losses perform well on specific metrics, the ensemble provides more consistent results across all classes, particularly for underrepresented tissues like NEC and INF. Visual examples in Figure 3 highlight the ensemble's segmentation quality and uncertainty estimation, especially around complex boundaries and rare regions.

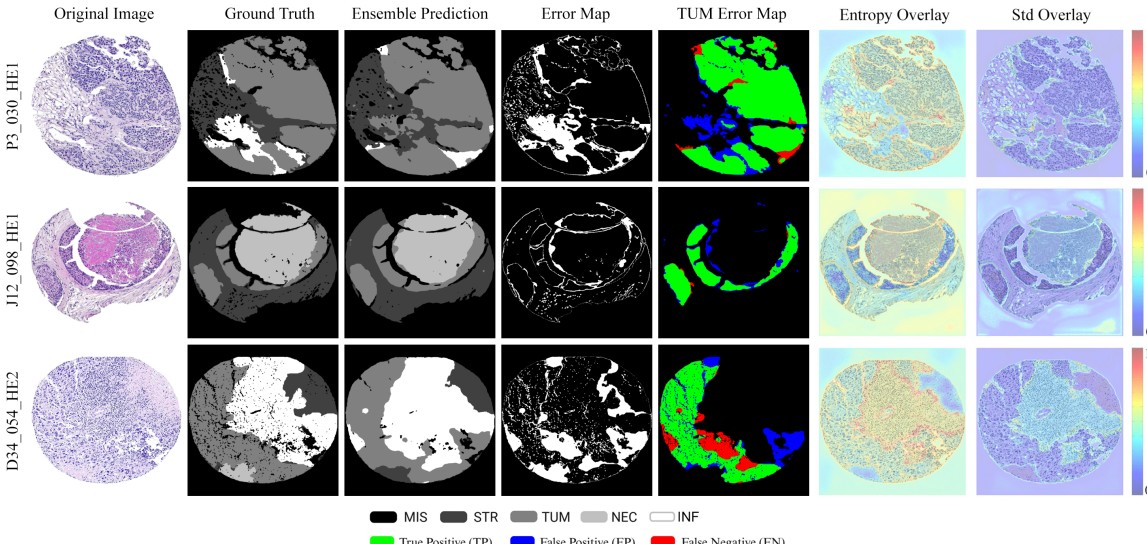

Figure 3: **Segmentation and uncertainty maps from the ensemble with MCDO.** Left to right: images, ground truth, predicted masks, and error maps (white = error); TUM-specific TP, FP, FN overlays; Entropy and STD overlays reveal high uncertainty at boundaries and minority classes.

### 3.3 Uncertainty Estimation

The MCDO-based uncertainty maps indicate higher prediction variability in underrepresented classes such as NEC and INF, and around tissue boundaries, as reflected by elevated STD and entropy values (Fig. 3). These observations suggest that MCDO effectively captures ambiguity in clinically relevant and structurally complex regions.

To evaluate the reliability of these estimates, we computed the UCE in addition to reliability diagrams (see Appendix D, Fig. 4). STD-based uncertainty consistently aligned better with actual error rates, closely following the ideal calibration diagonal. In contrast, entropy-based uncertainty tended to be misaligned, often overestimating confidence in low-error areas and underestimating it in high-error regions. This was quantitatively supported

by a substantially lower UCE for STD-based maps (0.085±0.033) compared to entropy-based ones $(0.404 \pm 0.085)$.

## 4 Discussion

Semantic segmentation of histopathological tissues remains a complex task due to blurred tissue boundaries and inter-observer variability. This study presents an automated pipeline for breast cancer TMA segmentation using an ensemble of U-Net models with ResNet encoders and MCDO for uncertainty estimation. Our method achieves a mean IoU of $0.58 \pm 0.11$ and Dice Score of $0.66 \pm 0.10$, outperforming similar algorithms like WeGleNet and HistoSegNet (Silva-Rodríguez et al. (2021); Chan et al. (2019)).

While IoU is a demanding metric—particularly for minority classes such as NEC and INF—our results demonstrate that combining it with the Dice Score offers a more comprehensive assessment. This distinction is particularly relevant in histopathology, where even small discrepancies can significantly impact the IoU. For example, minor errors in underrepresented classes can lead to a substantial drop in IoU, even when the segmentations appear visually acceptable.

A key contribution is the use of MCDO to estimate epistemic uncertainty, particularly in ambiguous regions and minority classes. Calibration analysis showed that standard deviation-based uncertainty more accurately reflected prediction errors (UCE = $0.085 \pm 0.033$) compared to entropy-based estimates (UCE = $0.404 \pm 0.085$). This demonstrates its value for enhancing model reliability and supporting clinical interpretability.

Nonetheless, the method has certain limitations. Ensemble models and MCDO increase computational overhead and may reduce predictive accuracy due to the use of dropout during inference, which poses challenges for clinical deployment. Future work should explore more advanced architectures, such as Transformers (Vaswani et al. (2017)) and posterior sampling techniques, to improve inference speed without compromising performance. Moreover, addressing class imbalance via sampling strategies or synthetic data augmentation could enhance the representation of minority classes.

Expanding the pipeline to incorporate multi-modal imaging (e.g., IHC or fluorescence), as well as to Whole Slide Imaging (WSI) could enable richer characterization of the TME, advancing both diagnostic accuracy and biological insights.

## 5 Conclusion

We propose a robust and interpretable pipeline for segmenting breast cancer tissue microarrays using U-Net ensembles with ResNet encoders, diverse loss functions, and MCDO for uncertainty estimation. Our method outperforms existing baselines and generalizes well across clinically relevant tissue types.

By incorporating uncertainty quantification, the approach improves reliability in ambiguous and rare regions, with strong calibration against prediction errors. Despite added computational cost, the framework lays the groundwork for future extensions to whole-slide and multi-modal histopathology, supporting biomarker discovery and advancing clinical decision support.

## Acknowledgments and Disclosure of Funding

This work was partially supported under grant PID2023-152631OB-I00 by the Ministerio de Ciencia, Innovación y Universidades, Agencia Estatal de Investigación (MCIN/AEI/10.13039/501100011033/), co-financed by European Regional Development Fund (ERDF), 'A way of making Europe'. L. Schmidt-Santiago is supported by the "Ayuda para contratación del ayudante de investigación (Programa de Empleo Juvelil PEJ-2023-AI/COM-27472)" project. R. Kinakh and S. Carrera-Salinas are supported under UC3M PIPF "Inteligencia Artificial, CONVOCATORIA 2024/D/DE/TD/1" PhD fellowship.

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

## Supplementary Material

## Appendix A. Loss Function Definitions

The following equations detail the five loss functions used in model training:

### A.1 Cross-Entropy (CE) Loss

A standard classification loss defined as:

$$\text{CE}(y, \hat{y}) = -\sum_{i=1}^{N} y_i \log(\hat{y}_i),$$

where $y_i \in \{0, 1\}$ is the ground truth label, $\hat{y}_i \in [0, 1]$ is the predicted probability, and $N$ is the total number of pixels.

### A.2 Dice Loss

Maximizes overlap between predicted and ground truth regions:

$$\text{Dice}(y, \hat{y}) = 1 - \frac{2\sum_{i=1}^{N} y_i \hat{y}_i}{\sum_{i=1}^{N} y_i + \sum_{i=1}^{N} \hat{y}_i}.$$

### A.3 Focal Loss

Focuses learning on hard examples:

$$\text{Focal}(y, \hat{y}) = -\sum_{i=1}^{N} (1 - \hat{y}_i)^{\gamma} y_i \log(\hat{y}_i),$$

where $\gamma > 0$ is a focusing parameter.

### A.4 Focal Dice Loss

Combines the benefits of Focal and Dice Loss:

$$\text{FocalDice}(y, \hat{y}) = 1 - \frac{2\sum_{i=1}^{N} (1 - \hat{y}_i)^{\gamma} y_i \hat{y}_i}{\sum_{i=1}^{N} y_i + \sum_{i=1}^{N} \hat{y}_i}.$$

**A.5 Tversky Loss**

A generalization of Dice with tunable false positive/negative weighting:

$$\text{Tversky}(y, \hat{y}) = 1 - \frac{\sum_{i=1}^{N} y_i \hat{y}_i}{\sum_{i=1}^{N} y_i \hat{y}_i + \alpha \sum_{i=1}^{N} y_i(1 - \hat{y}_i) + \beta \sum_{i=1}^{N}(1 - y_i)\hat{y}_i},$$

where $\alpha$ and $\beta$ control the weighting of false negatives and false positives, respectively.

## Appendix B. Uncertainty Metrics

**Standard Deviation (STD)**

Given $T$ stochastic forward passes, the standard deviation at pixel $i$ is computed as:

$$\sigma_i = \sqrt{\frac{1}{T} \sum_{t=1}^{T} (p_{i,t} - \bar{p}_i)^2},$$

where $p_{i,t}$ is the predicted probability at iteration $t$, and $\bar{p}_i$ is the mean probability across $T$ passes.

## Appendix C. Evaluation Metrics

The following metrics are used to assess segmentation performance:

- **Intersection over Union (IoU):** $\text{IoU}_c = \frac{|y_c \cap \hat{y}_c|}{|y_c \cup \hat{y}_c|}$

- **Dice Score:** $\text{Dice}_c = \frac{2|y_c \cap \hat{y}_c|}{|y_c| + |\hat{y}_c|}$

- **Accuracy:** $\text{Accuracy} = \frac{\text{TP} + \text{TN}}{\text{TP} + \text{TN} + \text{FP} + \text{FN}}$

- **Precision:** $\text{Precision}_c = \frac{\text{TP}_c}{\text{TP}_c + \text{FP}_c}$

- **Recall:** $\text{Recall}_c = \frac{\text{TP}_c}{\text{TP}_c + \text{FN}_c}$

The mean for each metric is calculated by averaging across all classes.

## Appendix D. Supplementary Figures

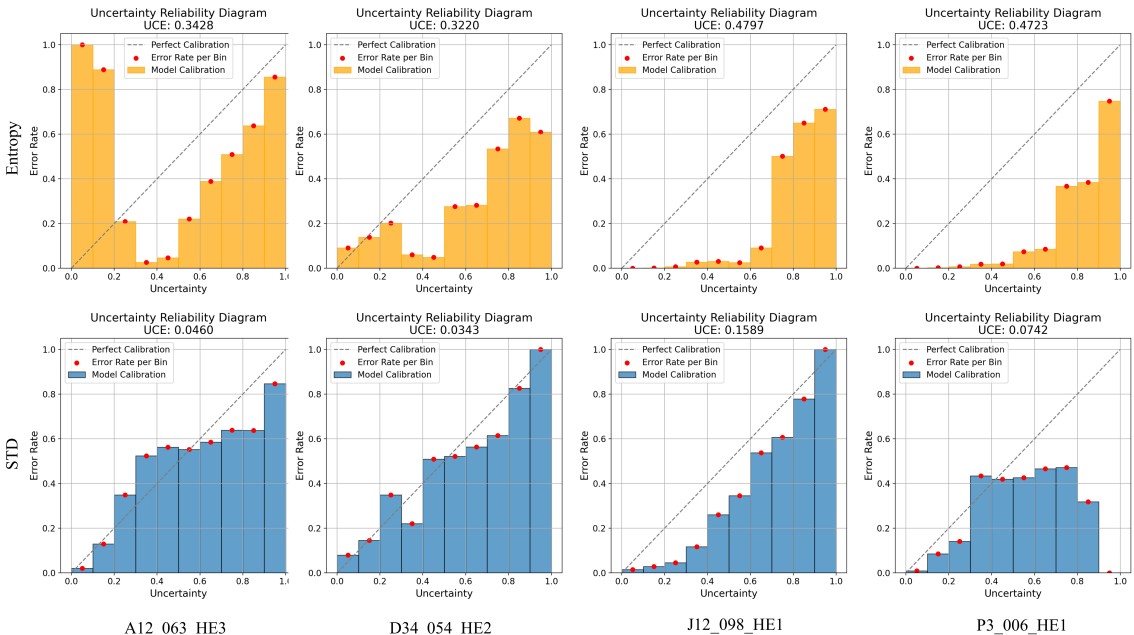

Figure 4: **Reliability diagrams for uncertainty calibration.** Comparison between predicted uncertainty and actual error rates across entropy-based and STD-based uncertainty maps. STD-based uncertainty aligns more closely with the diagonal, indicating better calibration. Entropy-based estimates show overconfidence in low-error areas and underconfidence in ambiguous regions.

