# OpenReview forum: "Uncertainty-Aware Ensemble Segmentation of Breast Cancer Tissue Microarrays"
_MICCAI.org/2025/Workshop/COMPAYL — COMPAYL 2025_

### Official Review · Reviewer_HFbD · 2025-07-11
**Well-written paper with potential experimental setup flaw**

**Rating:** 3
**Confidence:** 5

**Review:**

This is a well-written and complete paper on a topic that is still relevant (tissue segmentation in histopathology). While the work has a rather limited scope to only TMAs of BC, the resulting model might still prove useful for downstream analysis tasks. However, I have one major and some minor remarks that need to be addressed:

Major remarks:
The dataset contains three TMA cores per patient. 123 cores were annotated, of which 110 went into cross-validation and 13 into the final test set . The manuscript never states that all cores from the same patient were kept together within a fold or confined to either train or test. Without that guarantee the reported numbers can be inflated by information leakage across related cores. Unless this is clarified and/or addressed, the manuscript should not be accepted in my opinion.

Minor remarks:
Only tumour, necrosis and inflammation were traced by hand and everything else becomes an automatically generated MIS mask. The authors do not explain why these three compartments were prioritised. A short justification would strengthen the clinical relevance of the study.

For the frequency relevance weighted IoU the paper fixes the weights at 0.50 for tumour, 0.15 for stroma, necrosis and inflammation and 0.05 for miscellaneous . No empirical or clinical basis is provided, nor is a sensitivity analysis offered, so this is very arbitrary choice.

---

### Official Review · Reviewer_Ckzb · 2025-07-11
**Interesting paper, some clarification needed**

**Rating:** 4
**Confidence:** 4

**Review:**

This paper proposes a method for segmenting various tissue types within breast cancer tissue microarrays (TMAs): tumor epithelium, tumor-associated stroma, necrosis, and inflammation. A pre-processing pipeline removes non-cellular and disrupted tissue, followed by a segmentation approach that uses an ensemble of five U-Net models, each trained on a different loss function. At inference time, soft voting combines the model predictions. Stain normalization and Monte Carlo dropout (MCDO) are incorporated and evaluated via ablation. MCDO is also used to generate pixel-wise uncertainty maps via standard deviation and entropy.

**Strengths**
- Clear description of the biological target classes; visual examples (e.g., in a presentation setting) would enhance this further.
- Methodologically interesting approach.
- Uncertainty quantification is well-motivated, with an insightful evaluation.
- Study design is appropriate: a separate test set is held out, and ablation studies for novel additions are conducted and their results reported.
- Includes comparison to existing segmentation methods (WeGleNet and HistoSegNet), which helps interpret the usefulness of the performance.

**Weaknesses**
- Data selection: It is unclear how the 123 annotated cores were selected from the original pool of 4,944. Was this a random sample? Were they enriched for certain morphologies?
- Ground truth quality: The pathologist only selected regions of interest, while segmentation was performed by a trained assistant without expert review. This raises concerns about annotation reliability, especially for difficult or underrepresented classes like inflammation.
- Image resolution and resizing: The original scanning resolution is not reported. The images were resized to 1024×1024, but it is unclear whether this was done in a way that preserved spatial resolution across samples, or whether any scaling artifacts were introduced. This may have impacted performance.
- Evaluation metrics: The frequency-relevance weights used for frw-IoU are not justified or referenced. Clarifying whether these were based on clinical input or empirical tuning would be helpful.
- Failure analysis: Figure 3 effectively shows success cases and uncertainty overlays, but a version showing failure cases or significant misclassifications would strengthen the interpretation and transparency of results.

This is a solid paper with methodological novelty and appropriate evaluation, though some methodological details require clarification.